# Cocoa Shell Extract Reduces Blood Pressure in Aged Hypertensive Rats via the Cardiovascular Upregulation of Endothelial Nitric Oxide Synthase and Nuclear Factor (Erythroid-Derived 2)-like 2 Protein Expression

**DOI:** 10.3390/antiox12091698

**Published:** 2023-08-31

**Authors:** Santiago Ruvira, Pilar Rodríguez-Rodríguez, David Ramiro-Cortijo, María Martín-Trueba, María A. Martín-Cabrejas, Silvia M. Arribas

**Affiliations:** 1Department of Physiology, Faculty of Medicine, Universidad Autónoma de Madrid, C/Arbobispo Morcillo 2, 28029 Madrid, Spain; 2Food, Oxidative Stress and Cardiovascular Health (FOSCH) Research Group, Universidad Autónoma de Madrid, Ciudad Universitaria de Cantoblanco, 28049 Madrid, Spain; 3Ph.D. Program in Pharmacology and Physiology, Doctoral School, Universidad Autónoma de Madrid, C/Francisco Tomás y Valiente 2, 28049 Madrid, Spain; 4Institute of Food Science Research (CIAL), Universidad Autónoma de Madrid (UAM-CSIC), C/Nicolás Cabrera 9, 28049 Madrid, Spain; 5Department of Agricultural Chemistry and Food Science, Faculty of Science, Universidad Autónoma de Madrid, Ciudad Universitaria de Cantoblanco, 28049 Madrid, Spain

**Keywords:** aging, cocoa shell, food by-products, hypertension, Nrf2, e-NOS, GSH, maternal undernutrition

## Abstract

Cocoa shell is a by-product of cocoa manufacturing. We obtained an aqueous extract (CSE) rich in polyphenols and methylxanthines with antioxidant and vasodilatory properties. We aimed to evaluate the effects of CSE supplementation in aged hypertensive rats on blood pressure and the mechanism implicated. Eighteen-month-old male and female rats exposed to undernutrition during the fetal period who developed hypertension, with a milder form in females, were used (MUN rats). Systolic blood pressure (SBP; tail-cuff plethysmography) and a blood sample were obtained before (basal) and after CSE supplementation (250 mg/kg; 2 weeks, 5 days/week). Plasma SOD, catalase activity, GSH, carbonyls, and lipid peroxidation were assessed (spectrophotometry). In hearts and aortas from supplemented and non-supplemented age-matched rats, we evaluated the protein expression of SOD-2, catalase, HO-1, UCP-2, total and phosphorylated Nrf2 and e-NOS (Western blot), and aorta media thickness (confocal microscopy). MUN males had higher SBP compared with females, which was reduced via CSE supplementation with a significant difference for group, sex, and interaction effect. After supplementation with plasma, GSH, but not catalase or SOD, was elevated in males and females. Compared with non-supplemented rats, CSE-supplemented males and females exhibited increased aorta e-NOS and Nrf2 protein expression and cardiac phosphorylated-Nrf2, without changes in SOD-2, catalase, HO-1, or UCP-2 in cardiovascular tissues or aorta remodeling. In conclusion, CSE supplementation induces antihypertensive actions related to the upregulation of e-NOS and Nrf2 expression and GSH elevation and a possible direct antioxidant effect of CSE bioactive components. Two weeks of supplementation may be insufficient to increase antioxidant enzyme expression.

## 1. Introduction

Cardiovascular diseases (CVD) are collectively one of the leading causes of morbidity and mortality and are an important contributor to the burden for health systems worldwide. Hypertension is the main modifiable risk factor of CVD, which remains very prevalent, particularly in aged populations, despite pharmacological efforts to control it. To address the risks of high blood pressure to health, a multilevel approach, including pharmacological and nonpharmacological interventions, has been proposed [1]. Oxidative stress is one of the key hallmarks associated with hypertension and a relevant mechanism in CVD through the impairment of vasodilatation, and it promotes cardiovascular fibrosis and remodeling [2]. In this context, the use of foods or food ingredients rich in antioxidants is gradually gaining interest and is a current focus of research [3]. In fact, numerous epidemiological studies and clinical trials have evidenced the favorable effects of plant-based foods with high antioxidant contents in ameliorating CVD [4,5,6]. Among the bioactive components, phenolic compounds have gained interest, and a recent meta-analysis confirmed that their intake is inversely associated with CVD [7], which also exhibits blood-pressure-reducing effects [8].

Food waste is a rising environmental problem, and it is of interest to recycle its by-products. Cocoa (*Theobroma cacao* L.) processing produces a significant amount of waste, with 90% of cocoa fruit’s total weight being discarded [9]. The global cocoa market size is expanding, and it is estimated to reach EUR 13,795 M by 2027, a volume with detrimental effects on the environment [10]. Some cocoa industry by-products have an interesting composition in their bioactive molecules, including phenolic compounds, theobromine, and caffeine. This has promoted an interest in their revalorization for applications beyond traditional uses (fertilizer or feedstuff), including developing novel, high-value products such as nutraceuticals or functional food ingredients [10,11]. Among cocoa by-products, the cocoa shell, produced during cocoa bean roasting, is of potential interest. Our research group investigated this by-product and obtained an aqueous extract (CSE) with high levels of phenolic compounds, caffeine, and theobromine [12]. We also demonstrated that CSE exhibits antioxidant properties, improving mitochondrial function in cultured cells [13,14]. The antioxidant properties of CSE have also been shown in vascular tissue, showing that it is able to effectively scavenge superoxide anions, and in arteries from aged rats with endothelial dysfunction, CSE incubation improves vasodilatation, protecting nitric oxide (NO) from degradation [15]. In these studies, we also demonstrated that caffeine and phenolic components are present in CSE, accounting for its antioxidant and vasoactive effects. This is in line with recent meta-analyses and randomized control clinical trials, which have shown the relationship between the consumption of cocoa and cocoa flavanols and the improvement of endothelial function [8,16,17].

Based on previously demonstrated antioxidant and vasoactive properties of CSE *in vitro*, its relevant bioactive molecule content, interest in recycling cocoa manufacturing by-products, and the relevance of hypertension, our aim was to evaluate the capacity of a supplement based on CSE to reduce blood pressure in a rat model of hypertension. 

## 2. Materials and Methods

### 2.1. Experimental Model of Hypertension Induced by Fetal Undernutrition

Experiments were performed on 18-month-old male and female Sprague Dawley rats from a colony maintained at the Animal House Facility of the Universidad Autónoma de Madrid (ES-28079-0000097). The rats and experimental procedures were approved by the Ethics Review Board of the Universidad Autónoma de Madrid and by the Regional Environment Committee of the Comunidad Autónoma de Madrid (RD 53/2013; Ref. PROEX 04/19, Madrid, Spain).

The experimental model of hypertension was induced via exposure to undernutrition during fetal life through maternal undernutrition during gestation (MUN rat model). This intervention induces hypertension in males at the age of 6 months, while females exhibit a mild increase in blood pressure only during aging [18,19]. Briefly, after evidence of gestation shown by the presence of a vaginal plug, the dams are fed ad libitum during the first 10 days of gestation, and food is restricted to 50% of the usual rat daily intake from day 11 to delivery (12 g/day), returning to an ad libitum diet during lactation. Drinking water is always provided ad libitum. At birth, the offspring are standardized to 12 individuals, including males and females, and are followed at different time points. In the present study, we used 10 males and 10 females from 4 different litters at the age of 18 months. 

Rats from the same sex were housed with poplar bedding in type III or type IV cages according to their weight in groups of 3–5 per box; maintained under controlled conditions (12/12 light/dark photoperiod, 22 °C, 40% relative humidity); and fed with a standard diet (Euro Rodent Diet 22; 5LF5, Labdiet, Madrid, Spain) containing 55% carbohydrates, 22% protein, 4.4% fat, 4.1% fiber, and 5.4% mineral. Water was provided ad libitum. Rat health and welfare were regularly monitored by the staff at the Animal House Facility. 

### 2.2. Experimental Design

Blood pressure was evaluated using tail-cuff plethysmography (see protocol below), and at the end of the third measurement, blood was collected from the tail in tubes containing 5% heparin. Thereafter, the rats were supplemented with CSE (250 mg/kg/day, for 2 weeks, 5 days/week). The supplementation was based on voluntary ingestion using a gelatin cube containing the exact dose of CSE based on the weight of the rat (see protocol below). At the end of the supplementation period, blood pressure was measured again, and blood was collected. The rats were sacrificed with CO_2_ exposure followed by exsanguination, and thoracic aortas and hearts were dissected. From the aorta, a 1 cm length segment immediate to the aortic arch was fixed in 4% paraformaldehyde and stored. From the hearts, the atria were discarded. The remaining aorta and ventricles were snap-frozen and kept at −80 °C. In a group of age- and sex-matched MUN rats without supplementation, the same tissues were collected for comparison.

### 2.3. Blood Pressure Measurements

Systolic blood pressure (SBP) was assessed via tail-cuff plethysmography coupled with a pressure acquisition system (CIBERTEC Niprem 645, Madrid, Spain), as previously described [19]. The rats were first included in a chamber at 37 °C for 10 min to induce vasodilatation and placed inside a soft support in the darkness to reduce stress. The pressure cuff was placed around the tail, inflated to 220 mmHg and, after several pressure inflate-deflate cycles, data were registered.

The measurements were performed over 3 consecutive days, always by the same qualified researcher. The data from the first day were not included in the statistical analysis since it was the day of adaptation to the procedure.

### 2.4. Supplementation Protocol

CSE supplementation was performed using gelatin cubes and voluntary ingestion, as we previously described [20]. The cubes were prepared with 100% bovine gelatin dissolved in hot water at a concentration of 140 g/L, adding vanilla flavor (4.8 mL/L). At this point, the mix was transferred to a mold for solidification (neutral gelatin), or the CSE was added to ensure a dose of (250 mg/kg/day) based on the weight of each rat and then transferred to the mold to prepare 1 cm^3^ size cubes. Both neutral and CSE-gelatins were left at 4 °C for solidification, and the individual cubes were extracted and frozen until used. 

For voluntary ingestion, the rat was first trained with neutral gelatin cubes until full acceptance by placing the animal in an empty box without bedding with the cube for 2 h. This procedure was repeated daily until the cube was fully ingested, indicating acceptance (usually between 3 and 4 days). Then, supplementation was initiated with the CSE-containing cubes at a dose of 250 mg/kg/day. To adjust the dose, the rats were weighed weekly. The CSE gelatin cubes were given for 2 weeks, 5 days/week (excluding weekends). We used this dose and regime of administration since we previously demonstrated, through a metabolomic study, the presence of bioactive ingredients from CSE in rat plasma with only 4 days of supplementation at 250 mg/kg/day and a further increase after a total of 8 days with a similar 2-day cessation of supplementation during the weekend [15].

### 2.5. Plasma Antioxidants and Biomarkers of Oxidative Damage

The collected blood samples were centrifuged for 10 min (900 g at 4 °C), and the plasma was aliquoted and stored at −80 °C until use.

Protein content was assessed with a Bradford protein content assay (Bior-Rad, Pleasanton, CA, USA), and absorbance was measured in a microplate reader (Synergy HT Multi-Mode, BioTek, VT, USA) at 595 nm, using bovine serum albumin as standard.

Reduced glutathione (GSH) was assessed as previously described [21]. A fluorometric method based on a reaction with o-phthalaldehyde was used, and fluorescence was measured in a microplate reader at 360 ± 40 nm excitation and 460 ± 40 nm emission wavelengths. GSH concentration was expressed as μmol GSH/mg of protein in the sample.

Plasma catalase activity was assessed using a kit (catalase activity assay kit, Bioquochem, Gijón, Spain) according to the manufacturer’s instructions and expressed as U catalase/mg protein in the sample. Absorbance was read at 540 nm in a microplate reader. Superoxide dismutase (SOD) activity was also assessed using a kit (SOD Activity Assay kit, KB-03-011, Bioquochem, Gijón, Spain), according to the manufacturer’s instructions, as previously described, [22]. Absorbance was read at 450 nm in a microplate reader, and SOD activity was expressed as U SOD/mL in the sample.

Plasma carbonyl levels were evaluated as previously described [21] with a 2,4 dinitrophenylhydrazine assay adapted to a microplate reader using an extinction coefficient of 2,4-dinitrophenylhydrazine (ε = 22,000 M/cm). The absorbance was measured at 595 nm, and the data were expressed as nmol/mg protein in the sample.

Plasma lipid peroxidation was assessed using a kit (Lipid Peroxidation Assay kit, Bioquochem; Gijón, Spain) that analyzes malondialdehyde (MDA) and 4-hydroxy-trans-2-nonenal (HNE) concentrations, as previously described [21]. The experiments were performed according to the manufacturer’s protocol, and absorbance was measured in the microplate reader. MDA + HNE content was expressed as μM.

### 2.6. Western Blotting

The levels of protein expression were assessed via Western blotting, as previously described for the heart [18] and aorta [21]. The frozen aorta and ventricles were first homogenized with lysis buffer with the following composition: 0.42 mM NaCl, 1 mM Na_4_P_2_O_7_, 1 mM DTT, 20 mM HEPES, 20 mM NaF, 1 mM Na_3_VO_4_, 1 mM EDTA, 1 mM EGTA, 20% glycerol, 2 mM phenylmethylsulfonyl fluoride (PMSF), 1 µL/mL leupeptin, 1 µL/mL aprotinin, and 0.5 µL/mL N-alpha-p-tosyl-L-lysine chloromethyl ketone hydrochloride (TLCK-hydrochloride). Thereafter, the tissue was centrifuged (10,000 rpm, 4 °C, 10 min), and the supernatant was collected and stored at −80 °C. Bradford protein assay dye (Bio-Rad, Pleasanton, CA, USA) was used to assess protein content in the samples, measuring absorbance in the microplate reader at 595 nm. 

Primary antibodies were incubated overnight at 4 °C. After washing, the secondary antibodies (anti-rabbit or anti-mouse IgG, peroxidase-conjugated) were incubated for 90 min. The blots were washed and incubated in commercially enhanced chemiluminescence reagents (ECL Prime, Amersham Biosciences, UK), and the resulting bands were detected using a ChemiDoc XRS + Imaging System (Bio-Rad, USA). To ensure equal sample loading, all blots were re-incubated with glyceraldehyde 3-phosphate dehydrogenase (GADPH). Blots were quantified using the Image Lab 6.1 software (Bio-Rad, USA), and the expression levels of each protein were normalized with GADPH.

The antibodies we used, with their dilution, purchase company, molecular weight (MW), SDS-PAGE %, and protein loading quantity, are shown in Table 1. 

All the gels used in the study are shown in Appendix A.

### 2.7. Study of Aorta Structure using Confocal Microscopy

Two rings were cut from the PFA-fixed aortic segment and mounted directly on a slide provided with a small well filled with CITIFLUOR-AF mounting medium (Citifluor Ltd., London, UK). The rings were visualized with a laser scanning confocal microscope (Leica TCS SP2, Leica Microsystems, Barcelona, Spain) at excitation 488 nm/emission 500–560 nm, the wavelength at which elastic lamellae can be detected via autofluorescence, showing the limits of the arterial media. Single images were captured with a ×10 air objective, and the MetaMorph^TM^ image analysis software (version 6.2.6, Molecular Devices, Universal Image Corporation, London, UK) was used to quantify media thickness.

### 2.8. Statistical Analysis 

Statistical analysis was performed with the R software (version 3.6.0, 2018, R Core Team, Vienna, Austria) within the RStudio interface using the *rio*, *dplyr*, *compareGroups*, *ggpubr*, *devtools*, and *ggplot2* packages. Given the small sample size, data were expressed as median and interquartile range [Q1; Q3. Statistical analysis was performed with 2-way ANOVA, analyzing the effects of sex and group (basal vs. post-CSE or supplemented vs. non-supplemented), extracting the interaction effect between them. Significance level was established at a *p*-value < 0.05.

## 3. Results

### 3.1. Effect of CSE Supplementation on Systolic Blood Pressure in Aged MUN Rats

MUN females had significantly lower SBP compared with males, as shown by the sex effect. CSE supplementation reduced SBP in MUN males but not female rats, as shown by the significant *p*-values of the group and group and sex interactions (Table 2). These data indicate that the effect of CSE supplementation depends on sex.

### 3.2. Effect of CSE Supplementation on Plasma Oxidative Status in Aged MUN Rats

Plasma catalase and SOD activities were not modified by supplementation, neither in males nor in females (Figure 1A,B). However, plasma GSH levels were increased after CSE supplementation in both males and females (Figure 1C), as shown by the significant *p*-values of the group effect but not the sex or interaction effect. These data suggest the effect of CSE on GSH, but it was not different between sexes.

Supplementation did not modify plasma LPO, which was representative of oxidative damage to lipid (Figure 2A) or carbonyl groups, a biomarker of protein oxidation (Figure 2B), as shown by the non-significant *p*-values in the group. We detected higher LPO values in males, evidenced by a *p*-value near statistical significance for sex. These data indicate a higher level of oxidative damage to lipids in males.

### 3.3. Effect of CSE Supplementation on Nrf2 Protein Expression in Hearts and Aortas from Aged MUN Rats

In the hearts, there was no difference in total Nrf2 expression between MUN rats receiving CSE compared with non-supplemented rats, neither in males nor in females, as shown by the non-significant *p*-values for group, sex, and interaction between them (Figure 3A). However, phosphorylated-Nrf2 protein expression was significantly higher in CSE-supplemented rats, both males and females, evidenced by a significant group *p*-value for the group but not for sex or the interaction effect (Figure 3B). The relationship p-Nrf2/total Nrf2 was statistically significant between groups (*p*-value = 0.016) but not between sexes (*p*-value = 0.45) or the interaction effect (*p*-value = 0.45), suggesting the effect of CSE on active cardiac Nrf2 in both sexes.

In the aortas, the total Nrf2 expression was higher in rats receiving CSE, both in males and females, as shown by the significant *p*-value for the group but not for sex or the interaction effect (Figure 3C). However, phosphorylated-Nrf2 only increased in supplemented male rats, as shown by the significant *p*-values of sex and the interaction effect (Figure 3D). The relationship p-Nrf2/total Nrf2 was not statistically significant between groups (*p*-value = 0.27), and it was near statistical significance for sexes (*p*-value = 0.06) and the interaction (*p*-value = 0.06). These data suggest that, in aortas, CSE supplementation had a larger effect on active Nrf2 in male rats.

### 3.4. Effect of CSE Supplementation on e-NOS Protein Expression in Hearts and Aortas from Aged MUN Rats

In hearts, CSE supplementation did not modify total e-NOS expression in either males or females, as shown by the non-significant *p*-values for group, sex, and the interaction effect (Figure 4A). Phosphorylated e-NOS tended to increase in supplement groups with a *p*-value near statistical significance, with no sex effect (Figure 4B). 

In the aortas, the total e-NOS increased in supplemented rats, which was significant in the group but not in the sex or interaction effect (Figure 4C). In the rat aortas, it was not possible to obtain an adequate level of p-eNOS expression for quantitative analysis.

### 3.5. Effect of CSE Supplementation on Protein Expression of Antioxidant Enzymes in Heart and Aorta from Aged MUN Rats

In hearts, we did not find statistical differences in the expression levels of the antioxidant enzymes evaluated (catalase, SOD, HO-1, and UCP2), as shown by the non-significant *p*-values for the group, sex, and the interaction effects (Figure 5). These data indicate that CSE supplementation was not able to induce the expression of cardiac antioxidant enzymes.

Similarly, in the aortas, no statistical differences were detected in the expression of any of the evaluated antioxidant enzymes between rats receiving CSE and those without supplementation considering group, sex, or the interaction between them (Figure 6), indicating a lack of effect from CSE supplementation.

### 3.6. Effect of CSE Supplementation on Aorta Structure from Aged MUN rats

CSE supplementation did not modify the medial layer thickness of the aortas, as shown by the non-significant *p*-value for the group. The thickness was smaller in females, as shown by the significant *p*-value between the sexes (Figure 7). These data indicate that CSE supplementation was not able to reduce remodeling in either males or females.

## 4. Discussion

The present work aimed to evaluate the *in vivo* effects of the short-term supplementation of aged hypertensive rats with CSE, an extract obtained from cocoa shells (a by-product of the chocolate industry) with previously demonstrated antioxidant and vasodilatory effects *in vitro*. Our main findings were that CSE supplementation was able to reduce blood pressure in male MUN rats, likely because of an upregulation in the expression of e-NOS and Nrf2 in cardiovascular tissues and the elevation of GSH, although we cannot discard a direct antioxidant effect due to CSE bioactive components. Despite the significant increase in Nrf2 in hearts and aortas, we did not detect a significant upregulation in antioxidant enzyme expression in these tissues or an improvement in vascular remodeling, probably because of the short duration or insufficient dose of CSE supplementation in counteracting the severity of the damage in the aged hypertensive rats.

The chocolate manufacturing industry generates considerable amounts of waste yearly with negative environmental consequences, making the re-utilization of its by-products highly desirable [10]. Cocoa shell is one of them, produced during the roasting of cocoa beans in significant amounts, representing around 20% of the cocoa seed. Cocoa shell is a good source of insoluble dietary fiber and bioactive compounds, suitable for applications meant to generate novel foods and nutraceuticals with health benefits [11,23]. The aqueous extract used in the present study was previously optimized and characterized by our group, showing the presence of 15 phenolic compounds, the major ones being protocatechuic acid, gallic acid, procyanidin B_2_, (−)-epicatechin, (+)-catechin, and important amounts of methylxanthines (theobromine and caffeine) [12]. Furthermore, we recently demonstrated that the main phenolic compounds persisted after simulated gastrointestinal digestion, indicative of their capacity to reach plasma. We also observed that the digested fraction exhibited a free-radical scavenging capacity, counteracting oxidative stress in culture cells [24]. 

CVD continues to rise despite considerable progress in prevention and treatment [25] and is an important contributor to mortality and morbidity worldwide, particularly in the context of an aging population [1]. In view of the lack of efficacious therapies, the development of novel strategies to combat CVD is an important demand, such as using natural products, which is an important focus of research [26]. Foods or food ingredients with antioxidant properties are of particular interest since oxidative stress is one of the underlying mechanisms implicated in hypertension and CVD [27,28], which is also a hallmark of aging [29]. The composition of CSE and its antioxidant actions, demonstrated *in vitro*, in cell cultures, and in arteries [15,24], suggests that this extract is a suitable ingredient for targeting hypertension and associated CVD. As an experimental model, we used rats exposed to undernourishment conditions during fetal life, induced by maternal undernutrition during gestation (MUN rats). This animal model mimics an adverse fetal environment that leads to intrauterine growth restriction in humans, which is an important risk factor in the development of hypertension and CVD [30,31]. We characterized the MUN rat model and showed that male rats develop hypertension in adult life, while females show a mild form of hypertension during aging [19]. The MUN rats also exhibited the typical hallmarks of CVD, namely, endothelial dysfunction [21], aorta hypertrophy [32], and cardiac remodeling and dysfunction [18]. We used this rat model to test CSE actions since MUN rats are deficient in antioxidants (GSH, thiols, and SOD) from an early age and show oxidative damage in adult life [33]. MUN females are protected from CVD during the fertile period and, during aging, exhibit moderate alterations in endothelial dysfunction and increased blood pressure [21], as observed in humans. 

Since the MUN rat model of hypertension has been fully characterized and compared with controls, in the present study, we did not use control rats and only evaluated the effect of CSE on the MUN rats, comparing males and females. Our data demonstrated that short CSE supplementation reduced blood pressure in MUN males but not in females. We suggest that sex differences in the response to CSE may be related to the higher degree of hypertension and oxidative damage in MUN males. This was evidenced in the present study by the larger plasma carbonyl levels, a biomarker of oxidative damage to proteins. These results are in accordance with our previous data showing larger oxidative stress in MUN males throughout their lifespan, from weaning to aging, due to both antioxidant deficiency and the increased expression of ROS-producing enzymes in cardiovascular tissues [18,21,33]. Furthermore, the level of endothelial dysfunction was also higher in adult MUN males compared with females [21]. Therefore, a higher level of oxidative stress is likely to contribute to impaired vasodilatation and higher blood pressure in MUN male rats compared with females. 

The blood-pressure-lowering effects of CSE are likely due to the presence of bioactive components in CSE, rich in phenolic compounds and methylxanthines [12]. In humans, randomized controlled trials have shown an association between chocolate consumption, blood pressure reduction, and improved cardiovascular performance due to the presence of polyphenols, especially flavanols, which exert favorable antioxidant, anti-inflammatory, and vasodilatory effects [34,35]. It is worth mentioning that bioactive components are likely much lower in chocolate compared with its by-products, such as the cocoa shell [11], since fermentation markedly reduces their content [36], suggesting that cocoa-shell-derived ingredients could be more effective. Theobromine and caffeine content in CSE may also contribute to this effect. Clinical trials have demonstrated the beneficial effect of chocolate [36] and coffee [37] consumption against CVD. Even though caffeine has traditionally been considered with caution in the context of hypertension, recent data from treated and untreated hypertensive patients suggest that caffeine raises blood pressure in the short term, but the moderate and habitual consumption of coffee does not adversely affect blood pressure, highlighting the protective effects of the antioxidant components of coffee [38,39]. All these data suggest that the bioactive components of CSE (caffeine, theobromine, and phenolic compounds) may be responsible for the blood-pressure-lowering effects observed in MUN rats. 

Regarding the mechanism implicated in blood pressure reduction by CSE supplementation in our experimental model, direct or indirect actions from bioactive components of CSE are possible. Regarding direct effects, we have evidence suggesting that CSE bioactive components (at least caffeine and theobromine) can reach plasma after 7 days of supplementation at 250/mg/kg/day, as can the *in vitro* vasodilator action of CSE and the above-mentioned molecules through superoxide anion scavenging and NO protection from degradation [15]. We also demonstrated endothelial dysfunction in MUN rats, which was more prominent in males [21]. Therefore, the direct antioxidant action of the bioactive components of the cocoa shell may have contributed to maintaining higher NO bioavailability, improving vasodilatation and, in turn, reducing blood pressure. This mechanism could also explain a larger blood-pressure-lowering effect on MUN males, who had a higher level of superoxide anion production and endothelial dysfunction. In addition, caffeine could also have a direct vasodilator effect on vascular smooth muscle cells, acting as a competitive inhibitor of phosphodiesterase and accumulating cAMP [40].

A second possibility is the indirect effect of CSE components stimulating the expression of enzymes relevant to cardiovascular function. According to our data, one of the possible effects is the stimulation of e-NOS expression in the aorta, which may counteract the endothelial dysfunction we previously observed in MUN rats [21]. A recent review by Serreli and Deiana [41] highlighted the role of polyphenols as stimulators of NO production through an increase in the expression of e-NOS and the activation of the enzyme through phosphorylation, a key event in the release of moderate doses of NO. We showed that eNOS phosphorylation in Ser1177 (implicated in enzyme activation) was elevated in the hearts of rats supplemented with CSE. It was not possible to detect p-eNOS in the aortas because of technical difficulties. However, the pattern of activation is likely the same. The effect of polyphenols on e-NOS can be induced via the Akt pathway, as found in diabetic mice treated with the flavonoid morin, which improved endothelial-dependent relaxation responses via increases in aortic e-NOS expression [42]. A similar effect has been observed in the aortas of male hypertensive rats supplemented with protocatechuic acid for 12 weeks [43]. Although we did not explore the Akt pathway, the relevant content of this hydroxybenzoic acid in CSE supports the possibility of a similar mechanism of action. CSE also contains caffeine [12], which could participate in e-NOS activation, increasing intracellular calcium or enzyme expression. It would have been interesting to evaluate whether CSE supplementation could increase the level of aortic NO through fluorescent indicators such as DAF-2-DA. Unfortunately, this artery has a large level of elastin, which is fluorescent at the same wavelength, making such measurements inaccurate. This limitation could be overcome using resistance vessels.

Besides e-NOS expression and NO synthesis, CSE may also improve NO bioavailability through antioxidant action [8]. We explored the possibility that CSE indirectly stimulates the level of antioxidants through the Nrf2 signaling pathway and the activation of antioxidant response elements (AREs) since Nrf2 activators have emerged as a novel therapeutic tool for CVD [26]. Furthermore, we have evidence of a deficiency in this transcription factor in MUN adult males in association with hypertension [21]. In cardiovascular tissues, we showed an increase in Nrf2 due to CSE supplementation in both males and females. Several of the bioactive components from CSE, namely, protocatechuic acid, gallic acid, and flavan-3-ols, can participate in the observed increase. Protocatechuic acid and other hydroxybenzoic acids activate the Nrf2 signaling pathway [44], and in a rat model of myocardial infarction treatment with protocatechuic-acid-stimulated Nrf2, it increases GSH and several antioxidant enzymes [45]. Gallic acid, another component of CSE, has also been shown to increase Nrf2, with beneficial effects on cardiac hypertrophy [46]. Epicatechin also improves blood pressure in a rat model of hypertension through Nrf2 activation and the improvement of endothelial function [47]. Nrf2 activation could be also induced indirectly through NO [48,49]. In our rat model, this is possible since we showed that CSE supplementation increased the expression of this transcription factor in parallel with e-NOS expression in cardiovascular tissues. 

We evaluated those enzymes that have been reported in the literature to be stimulated by Nrf2, such as SOD, catalase, and HO-1 [8,26], and also the expression of UCP-2, a mitochondrial ubiquitous enzyme that ameliorates ROS production within the mitochondrion [50,51] and has been shown to be stimulated by cocoa flavan-3-ols [52,53]. Despite the fact that we evidenced an increase in Nrf2, we did not detect a significant upregulation in the expression of any of the studied enzymes. We suggest that this may be due to insufficient supplementation or dosing duration. In SHR rats, another model of hypertension, protocatechuic acid supplementation for 12 weeks significantly improved SOD and catalase [43]. CSE supplementation was not able to improve vascular remodeling, another hallmark of hypertension, which we evidenced in this rat model [19,32]. Again, we suggest that this is due to the insufficient duration of supplementation since other studies with long-term phenolic compound treatments were effective in reversing cardiovascular remodeling. For example, resveratrol improved remodeling, reducing smooth muscle cell proliferation [54,55]; gallic acid treatment for 60 days was able to reduce cardiac hypertrophy [46]; and epigallocatechin gallate treatment for 30 days also reduced cardiac hypertrophy and fibrosis [56]. 

Our CSE supplementation regime was not able to increase plasma SOD or catalase activities, but a significant elevation of GSH was observed. This increase may be related to Nrf2 upregulation, which is able to stimulate its synthesis through the activation of glutamate–cysteine ligase, the rate-limiting enzyme responsible for GSH synthesis [26]. Another possibility is an increase in NO, which stimulates GSH synthesis [57]. We suggest that GSH may also contribute to CSE blood-pressure-lowering effects since it plays a key function in maintaining intracellular redox balance, one of the most powerful endogenous antioxidants in the cardiovascular system [58], making GSH enhancement a possible therapeutic strategy in the context of cardiac damage [59]. GSH deficiency is also common in aging, which may participate in endothelial dysfunction and hypertension. In fact, there is evidence that supplementation with GSH precursors, glycine, and N-acetylcysteine can reduce blood pressure in aged subjects [60] and improve endothelial dysfunction, increasing flow-mediated dilatation in postmenopausal women [61]. The capacity of GSH precursors to ameliorate CVD has also been confirmed in aged animals, showing that supplementation improves cardiac function [62] and increases life span [63].

## 5. Conclusions

The present study demonstrates the capacity of CSE supplementation to reduce blood pressure in aged male hypertensive rats through the stimulation of e-NOS and Nrf2 expression and GSH, together with the possible direct antioxidant effect of circulating bioactive components, improving NO availability and vasodilatation. We did not observe an elevation in antioxidant enzymes, probably because of the short duration of the treatment or low doses. There was a sex-dependent effect for CSE on blood pressure reduction, with a larger effect on males despite a similar degree of enzyme expression, which may be related to the higher oxidative damage in MUN males. However, the influence of sex on the CSE response is an aspect to be considered in future clinical trials with this extract. In summary, our data support the beneficial effects of cocoa shell by-products in the context of hypertension and aging and the possibility of conducting a clinical trial to develop food ingredients or nutraceuticals for human use, and they also contribute to the circular economy. The adjustment of doses and the length of supplementation are aspects that deserve to be explored in future studies.

## Figures and Tables

**Figure 1 antioxidants-12-01698-f001:**
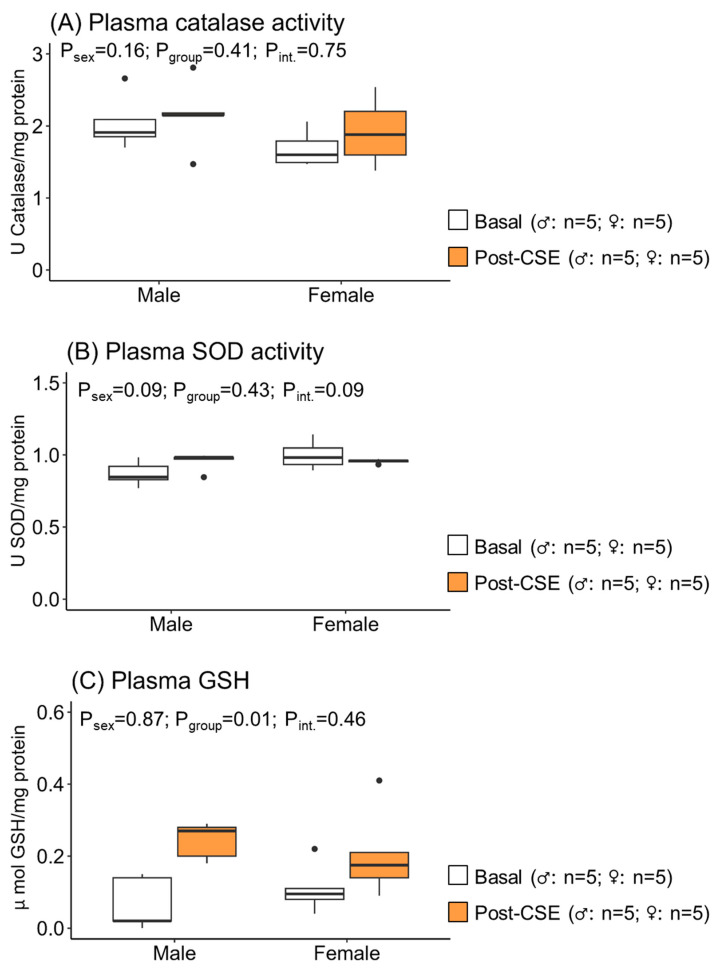
Plasma catalase activity (**A**), superoxide dismutase (SOD) activity (**B**), and GSH (**C**) in 18-month-old male and female maternal undernutrition (MUN) rats before (basal) and after supplementation with cocoa shell extract (250 mg/kg/day for 2 weeks, 5 days/week; post-CSE). Data are expressed as the median and interquartile range [Q1; Q3]. The *p*-value (P) was extracted with 2-way ANOVA considering sex and group factors and the interaction effect (int.); n indicates the sample size.

**Figure 2 antioxidants-12-01698-f002:**
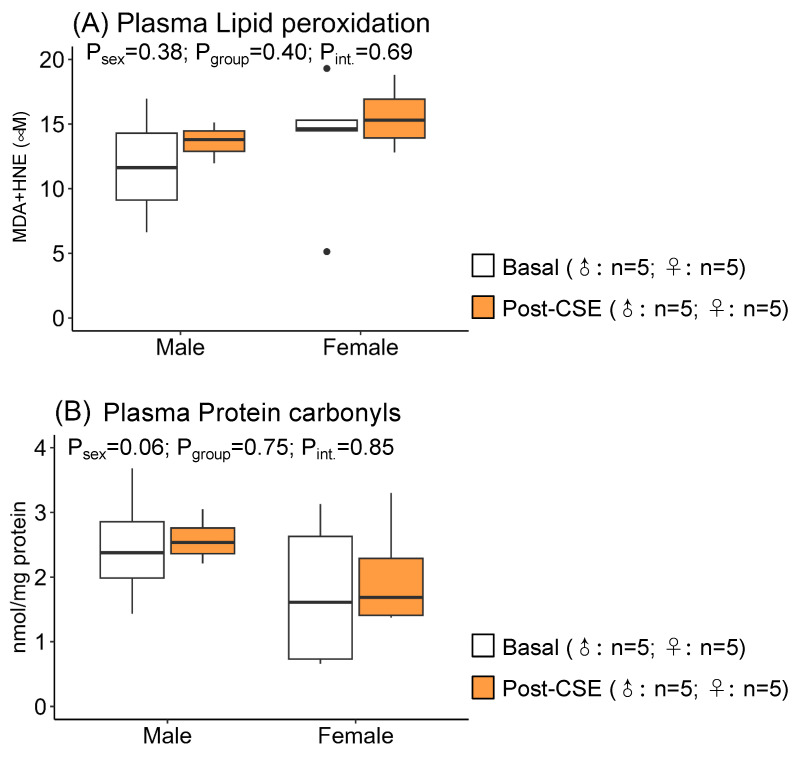
Plasma lipid peroxidation (**A**) and protein carbonyl (**B**) levels in male and female 18-month-old maternal undernutrition (MUN) rats before (basal) and after supplementation with cocoa shell extract (250 mg/kg/day for 2 weeks, 5 days/week; post-CSE). Data are expressed as median and interquartile range [Q1; Q3]. The *p*-value (P) was extracted with 2-way ANOVA considering sex and group factors and the interaction effect (int.); n indicates the sample size. MDA, malondialdehyde; HNE, 4-hydroxy-trans-2-nonenal.

**Figure 3 antioxidants-12-01698-f003:**
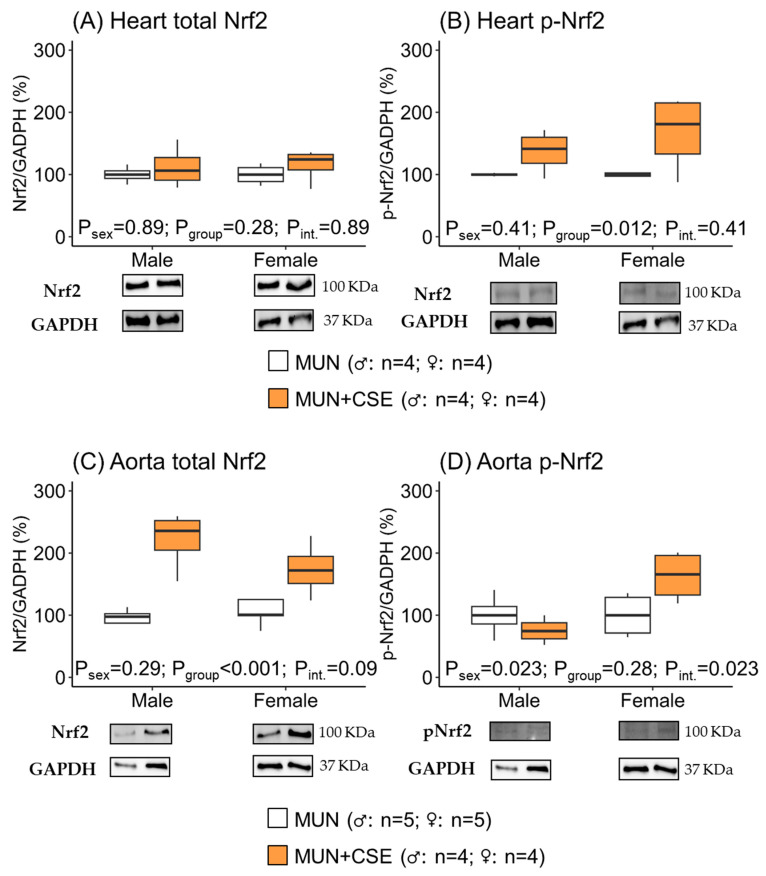
Protein expression levels of total Nrf2 and phosphorylated Nrf2 (p-Nrf2) in hearts (**A**,**B**) and aortas (**C**,**D**) from 18-month-old male and female maternal undernutrition (MUN) rats with and without supplementation with cocoa shell extract (250 mg/kg/day for 2 weeks, 5 days/week; MUN + CSE). Data are expressed as median and interquartile range [Q1; Q3]. The *p*-value (P) was extracted with 2-way ANOVA considering sex and group factors and the interaction effect (int.); n indicates the sample size. Nrf2, nuclear factor (erythroid-derived 2)-like 2; GADPH, glyceraldehyde phosphate dehydrogenase.

**Figure 4 antioxidants-12-01698-f004:**
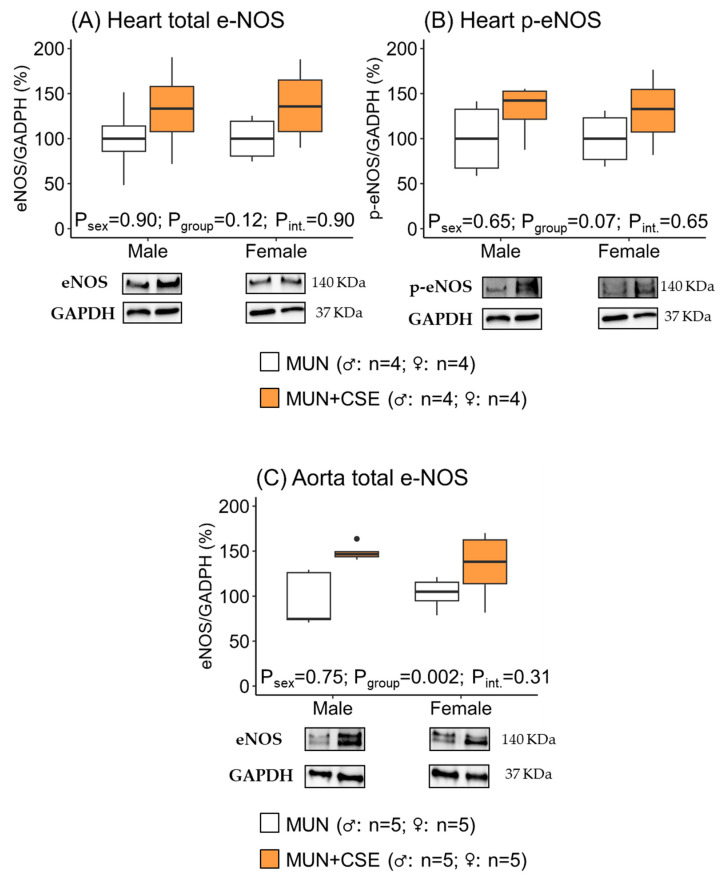
Protein expression levels of total e-NOS and phosphorylated e-NOS (p-e-NOS) in the hearts (**A**,**B**) and aortas (**C**) of 18-month-old male and female maternal undernutrition (MUN) rats with and without supplementation with cocoa shell extract (250 mg/kg/day for 2 weeks, 5 days/week; MUN + CSE). Data are expressed as median and interquartile range [Q1; Q3]. The *p*-value (P) was extracted with 2-way ANOVA considering sex and group factors and the interaction effect (int.); n, indicates the sample size. e-NOS, endothelial nitric oxide synthase; GADPH, glyceraldehyde phosphate dehydrogenase.

**Figure 5 antioxidants-12-01698-f005:**
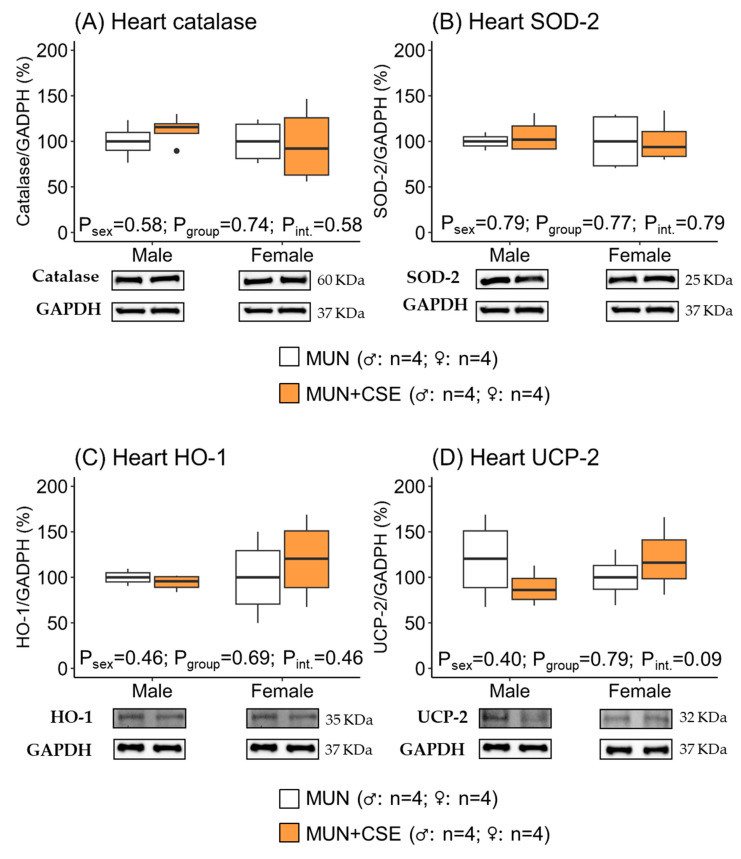
Protein expression levels of catalase (**A**), Mn-superoxide dismutase 2 (SOD-2; **B**), hemoxygenase-1 (HO-1; **C**), and uncoupling protein 2 (UPC-2; **D**) in hearts from 18-month-old male and female maternal undernutrition (MUN) rats with and without supplementation with cocoa shell extract (250 mg/kg/day for 2 weeks, 5 days/week; MUN + CSE). Data are expressed as median and interquartile range [Q1; Q3]. The *p*-value (P) was extracted with 2-way ANOVA considering sex and group factors and the interaction effect (int.); n indicates the sample size. SOD-2, Mn-superoxide dismutase; HO-1, hemoxygenase-1; UCP-2, uncoupling protein 2; GADPH, glyceraldehyde phosphate dehydrogenase.

**Figure 6 antioxidants-12-01698-f006:**
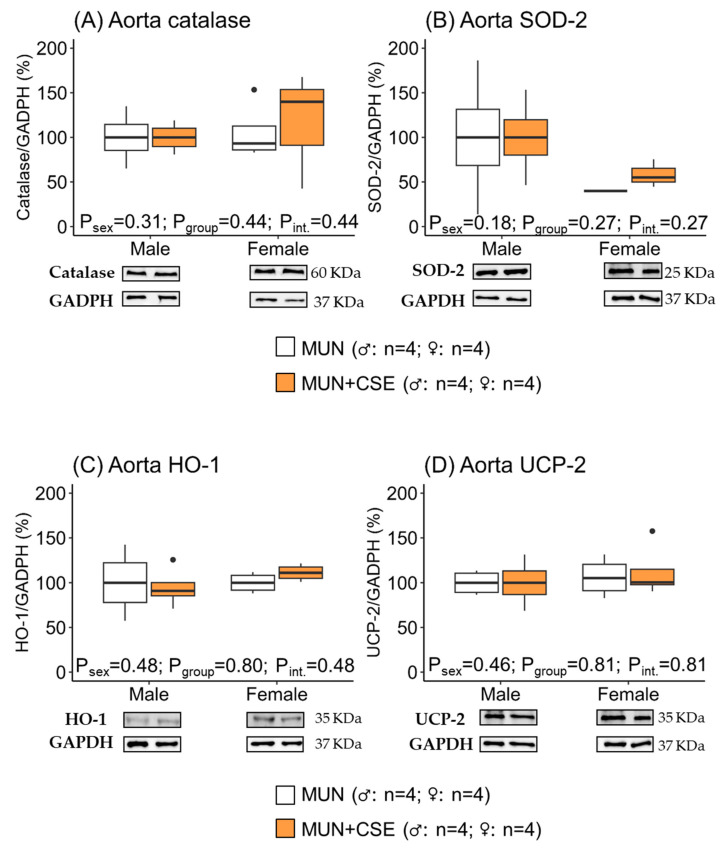
Protein expression levels of antioxidant enzymes in aortas from aged male and female MUN rats with and without CSE supplementation. Protein expression levels of catalase (**A**), superoxide dismutase 2 (**B**), hemoxygenase-1 (**C**), and uncoupling protein 2 (**D**) in aortas from 18-month-old male and female MUN rats with and without supplementation with cocoa shell extract (250 mg/kg/day for 2 weeks, 5 days/week; MUN + CSE). Data are expressed as median and interquartile range [Q1; Q3]. The *p*-value (P) was extracted with 2-way ANOVA considering sex and group factors and the interaction effect (int.); n indicates the sample size. SOD-2, Mn-superoxide dismutase; HO-1, hemoxygenase-1; UCP-2, uncoupling protein 2; GADPH, glyceraldehyde phosphate dehydrogenase; CSE, cocoa shell extract; MUN, maternal undernutrition.

**Figure 7 antioxidants-12-01698-f007:**
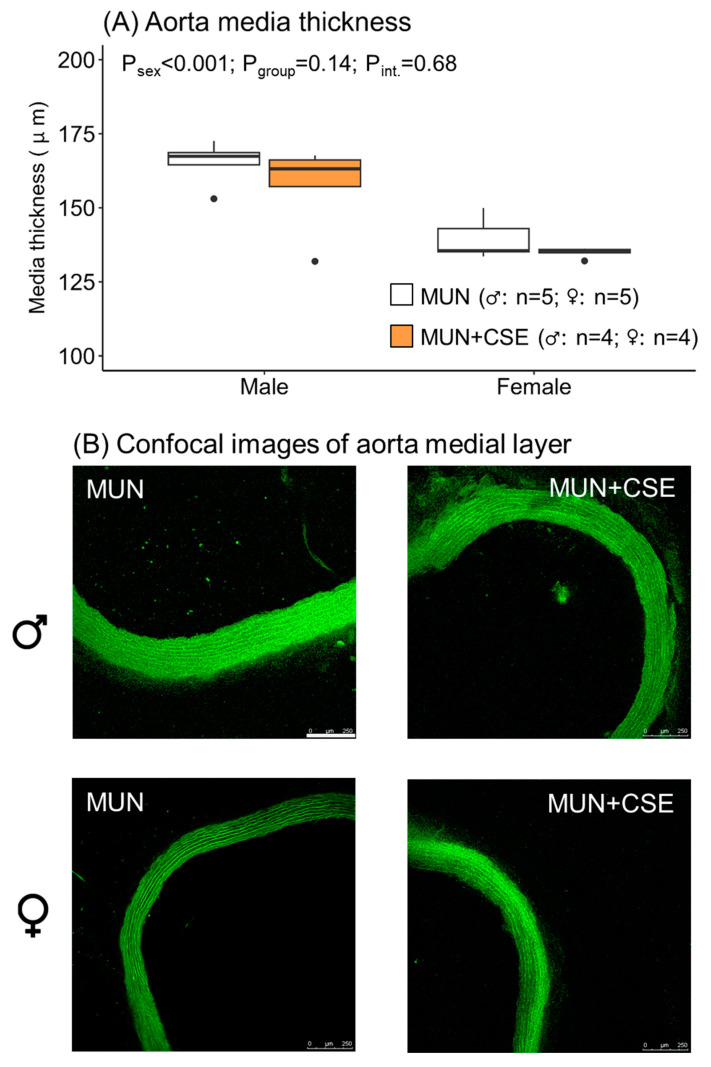
Media thickness (**A**) and representative confocal images (**B**) of the aorta from 18-month-old male and female MUN rats with and without CSE (250 mg/kg/day for 2 weeks, 5 days/week; MUN + CSE). Data are expressed as median and interquartile range [Q1; Q3]. The *p*-value (P) was extracted with 2-way ANOVA considering sex and group factors and the interaction effect (int.); n indicates the sample size. Images represent ring sections from the thoracic aorta taken with a spectral confocal microscope at Ex488 nm/Em500–560 nm with a ×10 air objective. Image bar, 250 μm.

**Table 1 antioxidants-12-01698-t001:** Antibodies and Western blotting conditions.

Antibody	Dilution	Company (City, Country)	MW (kDa)	Loading	%SDS-PAGE
e-NOS	1:250	BD Transduction (CA, USA)	140	30 μg	7%
p-eNOS (Ser 1177)	1:250	Cell Signaling Technology (MA, USA)	140	30 μg	7%
Nrf2	1:1000	Abcam (Cambridge, UK)	90	30 μg	12%
p-Nrf2 (Ser 40)	1:500	Thermo Fisher Scientific (MA, USA)	90	30 μg	12%
SOD-2	1:1000	Santa Cruz Biotech. (TX, USA)	25	30 μg	15%
Catalase	1:1000	Sigma-Aldrich (MO, USA)	60	30 μg	15%
HO-1	1:2000	Sigma Aldrich (MO, USA)	32	30 μg	12%
UCP-2	1:250	Santa Cruz Biotech. (TX, USA)	33	30 μg	12%
GADPH	1:5000	Cell Signalling Technology (MA, USA)	37	30 μg	Various

MW, molecular weight; e-NOS, endothelial nitric oxide synthase; p-eNOS, phosphorylated e-NOS; Nrf2, nuclear factor (erythroid-derived 2)-like 2; p-Nrf2, phosphorylated nuclear factor (erythroid-derived 2)-like 2; HO-1, hemoxygenase-1; SOD-2, superoxide dismutase-2 (Mn-SOD); UCP-2, uncoupling protein 2; GADPH, glyceraldehyde phosphate dehydrogenase.

**Table 2 antioxidants-12-01698-t002:** Modification of systolic blood pressure using CSE supplementation in aged male and female MUN rats.

Males (n = 5)		Females (n = 5)		*p*-Value
Basal (mmHg)	Post-CSE (mmHg)	Basal (mmHg)	Post-CSE (mmHg)	Group = 0.022
184.0 [180.0; 186.0]	159.0 [157.0; 162.0]	135.0 [132.0; 140.0]	140.0 [134.0; 149.0]	Sex = 0.010
				Interaction = 0.001

Systolic blood pressure measured via tail-cuff plethysmography in 18-month-old male and female MUN rats before (basal) and after supplementation with cocoa shell extract (250 mg/kg/day for 2 weeks, 5 days/week; post-CSE). The *p*-value was extracted with 2-way ANOVA considering sex and group factors and the interaction effect; n indicates the sample size.

## Data Availability

The data presented in this study are available upon request from the corresponding author. The availability of the data is restricted to investigators based in academic institutions.

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
