# Peer review of "Cocoa Shell Extract Reduces Blood Pressure in Aged Hypertensive Rats via the Cardiovascular Upregulation of Endothelial Nitric Oxide Synthase and Nuclear Factor (Erythroid-Derived 2)-like 2 Protein Expression"

_antioxidants, 2023, doi:10.3390/antiox12091698_

Round 1

Reviewer 1 Report

The authors aimed to evaluate capacity of a supplement based on cocoa shell aqueous extract (CSE) to reduce blood pressure in a rat model of hypertension. They concluded that  CSE supplementation induced antihypertensive actions, at least in part due to the increased expression of e-NOS and Nrf2 (resp. p-Nrf2) in the heart or aorta and plasma GSH.

The study is very interesting and beneficial, mainly because of the use of food waste which is a rising environmental problem. Moreover, such "waste" has been shown to have significant beneficial effects. I have only minor comments and questions.

Lines 110 and 114 refer to the protocol below. But the protocol is probably missing. What was the composition of the gelatin cube that contain the exact dose of CSE?

The data suggested that in the aorta CSE supplementation had a larger effect on active Nrf2 in male rats. The same is true for eNOS. But the basic values seem to be similar in male and female rats despite higher blood pressure in male rats. Can you briefly discuss this fact?

The significance (asterisks) is missing in the figure 3.

Figure 4 C: Total e-NOS seems to significantly increase only in the males. It should be written in the text.

no comments

Author Response

The authors aimed to evaluate capacity of a supplement based on cocoa shell aqueous extract (CSE) to reduce blood pressure in a rat model of hypertension. They concluded that CSE supplementation induced antihypertensive actions, at least in part due to the increased expression of e-NOS and Nrf2 (resp. p-Nrf2) in the heart or aorta and plasma GSH.

The study is very interesting and beneficial, mainly because of the use of food waste which is a rising environmental problem. Moreover, such "waste" has been shown to have significant beneficial effects. I have only minor comments and questions.

Answer: We would like to thank you for the comments and suggestions. We hope to have answered all of them and incorporated the changes in the text.

Lines 110 and 114 refer to the protocol below. But the protocol is probably missing. What was the composition of the gelatin cube that contain the exact dose of CSE?

Answer: You are right, the protocol was missing. A full description of the preparation of the gelatins and supplementation procedure is now included (lines 135-152).

The data suggested that in the aorta CSE supplementation had a larger effect on active Nrf2 in male rats. The same is true for eNOS. But the basic values seem to be similar in male and female rats despite higher blood pressure in male rats. Can you briefly discuss this fact?

Answer: Supplementation increased Nrf2 and e-NOS in both sexes, since there was no significant interaction between group and sex. It is true that it was near significance in Nrf2 (p-Value int=0.09). Although most basic parameters were quite similar between MUN males and females, oxidative damage to proteins (plasma carbonyls), tended to be larger in MUN males, shown (p-Value sex=0.06), which is in accordance with a higher level of oxidative stress in MUN males, as we have previously described in plasma (DOI: 10.1016/j.jnutbio.2015.08.004) and aorta (DOI: 10.1007/s13105-023-00949-1) and a higher expression of ROS-producing enzymes in heart (DOI: 10.1371/journal.pone.0171544). Also, the level of endothelial dysfunction is higher in adult MUN males compared to females (DOI: 10.1007/s13105-023-00949-1). All these alterations are likely to contribute to the higher level of blood pressure in MUN male rats compared to females. We have included this aspect in the discussion for clarification (lines 439-451). Additionally, we have found a mistake in figure 2B, and we have amended the title (plasma carbonyls instead of lipid peroxidation). 

The significance (asterisks) is missing in the figure 3.

Answer: There is no asterisk in any figure. The p-Value is sown within each figure. The points shown are not asterisks, they are outliers.

Figure 4C: Total e-NOS seems to significantly increase only in the males. It should be written in the text.

Answer: It is true that the trend seems to be found only in males. However, according to the Academic Editor suggestions, we performed a 2w-ANOVA analysis, which showed a significant difference in the group factor (MUN vs MUN+CSE) but not in the sex factor or their interaction, suggesting that the effect of CSE was not different between males and females.

Reviewer 2 Report

The authors in this manuscript evaluated the in vivo short-term (2 weeks, 5 days per week) supplementation of aged hypertensive rats with CSE. One of the main findings showed that CSE supplementation reduced blood pressure in male MUN rats, likely due to an up-regulation of the expression of eNOS and  Nrf2 in cardiovascular tissues and the elevation of GSH. Despite the significant increase of Nrf2 in heart and aorta, we did not detect a significant up-regulation of antioxidant enzyme expression in these tissues or an improvement in vascular remodeling. This is likely due to the short duration or insufficient dose of CSE supplementation to counteract the severity of the damage in aged hypertensive rats. Major comments for the manuscript are indicated below:

Major comments:

1. How is the dose and duration of CSE determined? Why are the animals only treated for 5 days per week, could this explain why there is no changes in the antioxidant enzymes or pathway? Perhaps there are insufficient level of CSE when the animals are killed for experiments.

2. What are the key bioactive components of CSE? How much of these bioactives are present in the animals after treatment?

3. It seems that there is a lack of aged normotensive group as control? Without this control group, it is unknown if there is any cardiovascular dysfunction induced by hypertension?

4. Given that antioxidant activity is one of the key of function of CSE, it would be important to measure ROS levels directly within the cardiovascular tissues. Measurement of antioxidant at plasma level is indirect.

5. CSE supplementation is shown to increase eNOS expression, which likely contribute to vasodilatory function and possibly reduction in blood pressure. Additional measurement of NO levels in the vascular tissues or functional measurement of endothelial function is required to support the western blot data.

6. Why the figures are presented as median and interquartile range? With n=5, it is still possible to present the data as mean +/-SEM, using scatter plot to demonstrate the variability in the sample group. This will be a better and meaningful method for data representation.

N/A

Author Response

The authors in this manuscript evaluated the in vivo short-term (2 weeks, 5 days per week) supplementation of aged hypertensive rats with CSE. One of the main findings showed that CSE supplementation reduced blood pressure in male MUN rats, likely due to an up-regulation of the expression of eNOS and Nrf2 in cardiovascular tissues and the elevation of GSH. Despite the significant increase of Nrf2 in heart and aorta, we did not detect a significant up-regulation of antioxidant enzyme expression in these tissues or an improvement in vascular remodeling. This is likely due to the short duration or insufficient dose of CSE supplementation to counteract the severity of the damage in aged hypertensive rats. Major comments for the manuscript are indicated below:

Answer:  We would like to thank you for the comments and suggestions. We have tried to provide explanation for them and include additional information in the text.

Major comments:

  1. How is the dose and duration of CSE determined? Why are the animals only treated for 5 days per week, could this explain why there is no changes in the antioxidant enzymes or pathway? Perhaps there are insufficient level of CSE when the animals are killed for experiments.

Answer: We decided on the dose based on our previous chronic toxicity studies with rodents with CSE (dose 1000 mg/kg/day). We considered appropriate to use ¼ of this dose to evaluate the capacity of the extract to exert biological effects. Besides, a preliminary metabolomic study demonstrated that after 4 days of CSE supplementation at 250 mg/kg/day, caffeine and theobromine are detectable in plasma, and after 7 days of supplementation (with a discontinuation during the weekend as in the present study) they are even higher (DOI: 10.3390/antiox11020429). That was also the reason for the supplementation regime of 5 days at 250 mg/kg/day. We have clarified this aspect in methods section (lines 145-152).

  1. What are the key bioactive components of CSE? How much of these bioactives are present in the animals after treatment?

Answer: As indicated above, using untargeted metabolomic analysis by LC-ESI-QTOF, we could detect a time-dependent increase in caffeine and theobromine in plasma of adult rats exposed to 250 mg/kg/day of supplementation for 1 week. Due to the method used, we could only detect methylxanthines and not phenolic compounds. However, we expect that their metabolites could also appear in plasma. We are now performing a full metabolomic analysis of the extract and the plasma of supplemented rats and expect to have additional information.

  1. It seems that there is a lack of aged normotensive group as control? Without this control group, it is unknown if there is any cardiovascular dysfunction induced by hypertension?

Answer: The MUN rat model of hypertension has been fully characterized and compared to control by our group in several studies and different age points, assessing blood pressure and cardiovascular function and structure. Thus, we have detected alterations in endothelial and heart function, most markedly in MUN males compared to their control (DOI: 10.1016/j.jnutbio.2015.08.004; DOI: 10.1007/s13105-023-00949-1; DOI: 10.1371/journal.pone.0171544). MUN females only evidenced vascular remodeling compared to their control and hypertension was detected in ageing only (DOI:10.3390/biomedicines8100424) and was not detected when the rats were anesthetized (DOI: 10.1371/journal.pone.0171544), suggesting a milder form of hypertension compared to males. Based on our previous knowledge on this model, in the present study our aim was to evaluate the effectiveness of the CSE supplementation on hypertension and the possible mechanisms implicated; therefore, we considered it was not necessary to use control rats and only evaluated the effect of CSE on MUN rats. Given the sex-differences we have previously detected, we also aimed to compare the possible differences between males and females in the response of CSE. We have included an explanation regarding these aspects in the discussion (lines 439-451).

  1. Given that antioxidant activity is one of the key of function of CSE, it would be important to measure ROS levels directly within the cardiovascular tissues. Measurement of antioxidant at plasma level is indirect.

Answer: We have previous evidence of increased superoxide anion production in arteries from MUN rats, higher in males (DOI: 10.1007/s13105-023-00949-1) and the in vitro capacity of CSE and its main components to effectively scavenge superoxide anion (DOI: 10.3390/antiox11020429). Besides, we have evidenced that the level of plasma oxidative status is related to the level of oxidative damage in arteries (DOI: 10.1007/s13105-023-00949-1) and heart (DOI: 10.1371/journal.pone.0171544). These aspects have been now better explained in the discussion section.

  1. CSE supplementation is shown to increase eNOS expression, which likely contribute to vasodilatory function and possibly reduction in blood pressure. Additional measurement of NO levels in the vascular tissues or functional measurement of endothelial function is required to support the western blot data.

Answer: You are right, this measurement would have been very informative. However, in the present study, we used the aortic tissue for western blotting analysis and, therefore, it is not possible to conduct such experiments. Besides, we have previously shown that assessment of NO by fluorescent indicators (such as DAF-2-DA) is not feasible in aorta (due to large autofluorescence coming from elastin) and that is why we did not conduct such experiments. We are aware of this limitation and indicated it in the discussion (lines 501-505). Given this important aspect, we are now performing new studies with CSE supplementation evaluating in resistance vessels NO release together with endothelial function.

  1. Why the figures are presented as median and interquartile range? With n=5, it is still possible to present the data as mean +/-SEM, using scatter plot to demonstrate the variability in the sample group. This will be a better and meaningful method for data representation.

Answer: We used the median and IQR since the recommendations are to use nonparametric analysis when the sample size is less than 30 (for one-sample) or 15/group (in sample comparisons) (https://statisticsbyjim.com/hypothesis-testing/nonparametric-parametric-tests/). Along with these conventions, we followed the recommendations of the Academic Editor who proposed a non-parametric strategy due to the sample size of the study, particularly for the western blotting data which were smaller. Accordingly, we used throughout the study this type of statistical analysis which are more robust techniques to detect significant differences when they truly exist.

Round 2

Reviewer 2 Report

I thank the reviewers for responding to my queries which are mostly appropriate. However, with question 2, it is mentioned that the authors are now performing a full metabolomic analysis of the extract and the plasma of supplemented rats and expect to have additional information. Will this information be included in the current manuscript?

N/A

Author Response

I thank the reviewers for responding to my queries which are mostly appropriate. However, with question 2, it is mentioned that the authors are now performing a full metabolomic analysis of the extract and the plasma of supplemented rats and expect to have additional information. Will this information be included in the current manuscript?

Answer: The authors are grateful for the time spent reviewing our work. We do not think providing the full metabolomic analysis in the present manuscript would be appropriate, since it is beyond the aim of the study, which is to evaluate the possible in vivo cardiovascular actions of the extract. Besides, the metabolomic analysis we are performing is not complete. It is a full chemometric evaluation to elucidate the plasmatic changes induced in rats after 8 days of CSE supplementation and the comparison with the native matrix. The information we provided in the current manuscript regarding the presence of bioactive components in plasma after supplementation (included in reference 15) was a preliminary study analyzing only the presence of methylxanthines, which enabled us to be confident that CSE bioactive components are reaching plasma even after short supplementation period.